# ON ROBUSTNESS OF NEURAL ORDINARY DIFFERENTIAL EQUATIONS

**Hanshu YAN\*, Jiawei DU\*, Vincent Y. F. TAN & Jiashi FENG**
Department of Electrical and Computer Engineering
National University of Singapore
`{hanshu.yan, dujiawei}@u.nus.edu, {vtan, elefjia}@nus.edu.sg`

## ABSTRACT

Neural ordinary differential equations (ODEs) have been attracting increasing attention in various research domains recently. There have been some works studying optimization issues and approximation capabilities of neural ODEs, but their robustness is still yet unclear. In this work, we fill this important gap by exploring robustness properties of neural ODEs both empirically and theoretically. We first present an empirical study on the robustness of the neural ODE-based networks (ODENets) by exposing them to inputs with various types of perturbations and subsequently investigating the changes of the corresponding outputs. In contrast to conventional convolutional neural networks (CNNs), we find that the ODENets are more robust against both random Gaussian perturbations and adversarial attack examples. We then provide an insightful understanding of this phenomenon by exploiting a certain desirable property of the flow of a continuous-time ODE, namely that integral curves are non-intersecting. Our work suggests that, due to their intrinsic robustness, it is promising to use neural ODEs as a basic block for building robust deep network models. To further enhance the robustness of vanilla neural ODEs, we propose the time-invariant steady neural ODE (TisODE), which regularizes the flow on perturbed data via the time-invariant property and the imposition of a steady-state constraint. We show that the TisODE method outperforms vanilla neural ODEs and also can work in conjunction with other state-of-the-art architectural methods to build more robust deep networks.

## 1 INTRODUCTION

Neural ordinary differential equations (Chen et al., 2018) form a family of models that approximate nonlinear mappings by using continuous-time ODEs. Due to their desirable properties, such as invertibility and parameter efficiency, neural ODEs have attracted increasing attention recently (Dupont et al., 2019; Liu et al., 2019). For example, Grathwohl et al. (2018) proposed a neural ODE-based generative model—the FFJORD—to solve inverse problems; Quaglino et al. (2019) used a higher-order approximation of the states in a neural ODE, and proposed the SNet to accelerate computation. Along with the wider deployment of neural ODEs, robustness issues come to the fore. However, the robustness of neural ODEs is still yet unclear. In particular, it is unclear how robust neural ODEs are in comparison to the widely-used CNNs. Robustness properties of CNNs have been studied extensively. In this work, we present the first systematic study on exploring the robustness properties of neural ODEs.

To do so, we consider the task of image classification. We expect that results would be similar for other machine learning tasks such as regression. Neural ODEs are dimension-preserving mappings, but a classification model transforms a high-dimensional input—such as an image—into an output whose dimension is equal to the number of classes. Thus, we consider the neural ODE-based classification network (ODENet) whose architecture is shown in Figure 1. An ODENet consists of three components: the feature extractor (FE) consists of convolutional layers which maps an input datum to a multi-channel feature map, a neural ODE that serves as the nonlinear representation mapping (RM), and the fully-connected classifier (FCC) that generates a prediction vector based on the output of the RM.

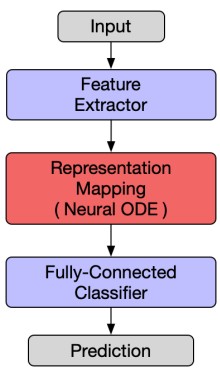

The robustness of a classification model can be evaluated through the lens of its performance on perturbed images. To comprehensively investigate the robustness of neural ODEs, we perturb original images with commonly-used perturbations, namely, random Gaussian noise (Szegedy et al., 2013) and harmful adversarial examples (Goodfellow et al., 2014; Madry et al., 2017). We conduct experiments in two common settings—training the model only on authentic non-perturbed images and training the model on authentic images as well as the Gaussian perturbed ones. We observe that ODENets are more robust compared to CNN models against all types of perturbations in both settings. We then provide an insightful understanding of such intriguing robustness of neural ODEs by exploiting a certain property of the flow (Dupont et al., 2019), namely that integral curves that start at distinct initial states are non-intersecting. The flow of a continuous-time ODE is defined as the family of solutions/paths traversed by the state, starting from different initial points, and an integral curve is a specific solution for a given initial point. The non-intersecting property indicates that an integral curve starting from some point is constrained by the integral curves starting from that point's neighborhood. Thus, in an ODENet, if a correctly classified datum is slightly perturbed, the integral curve associated to its perturbed version would not change too much from the original one. Consequently, the perturbed datum could still be correctly classified. Thus, there exists intrinsic robustness regularization in ODENets, which is absent from CNNs.

Figure 1: The architecture of an ODENet. The neural ODE block serves as a dimension-preserving nonlinear mapping.

Motivated by this property of the neural ODE flow, we attempt to explore a more robust neural ODE architecture by introducing stronger regularization on the flow. We thus propose a *Time-Invariant Steady neural ODE* (TisODE). The TisODE removes the time dependence of the dynamics in an ODE and imposes a steady-state constraint on the integral curves. Removing the time dependence of the derivative results in the time-invariant property of the ODE. To wit, given a solution $\mathbf{z}_1(t)$, another solution $\widetilde{\mathbf{z}}_1(t)$, with an initial state $\widetilde{\mathbf{z}}_1(0) = \mathbf{z}_1(T')$ for some $T' > 0$, can be regarded as the $-T'$- shift version of $\mathbf{z}_1(t)$. Such a time-invariant property would make bounding the difference between output states convenient. To elaborate, let the output of a neural ODE correspond to states at time $T > 0$. By the time-invariant property, the difference between outputs, $\|\widetilde{\mathbf{z}}_1(T) - \mathbf{z}_1(T)\|$, equals to $\|\mathbf{z}_1(T + T') - \mathbf{z}_1(T)\|$. To control this distance, a steady-state regularization term is introduced to the overall objective to constrain the change of a state after time exceeds $T$. With the time-invariant property and the steady-state term, we show that TisODE even is more robust. We do so by evaluating the robustness of TisODE-based classifiers against various types of perturbations and observe that such models are more robust than vanilla ODE-based models.

In addition, some other effective architectural solutions have also been recently proposed to improve the robustness of CNNs. For example, Xie et al. (2017) randomly resizes or pads zeros into test images to destroy the specific structure of adversarial perturbations. Besides, the model proposed by Xie et al. (2019) contains feature denoising filters to remove the feature-level patterns of adversarial examples. We conduct experiments to show that our proposed TisODE can work seamlessly and in conjunction with these methods to further boost the robustness of deep models. Thus, the proposed TisODE can be used as a generally applicable and effective component for improving the robustness of deep models.

In summary, our contributions are as follows. Firstly, we are the first to provide a systematic empirical study on the robustness of neural ODEs and find that the neural ODE-based models are more robust compared to conventional CNN models. This finding inspires new applications of neural ODEs in improving robustness of deep models, a problem that concerns many deep learning theorists and practitioners alike. Secondly, we propose the TisODE method, which is simple yet effective in significantly boosting the robustness of neural ODEs. Moreover, the proposed TisODE can also be used in conjunction with other state-of-the-art robust architectures. Thus, TisODE can serve as a drop-in module to improve the robustness of deep models effectively.

## 2 PRELIMINARIES ON NEURAL ODE

It has been shown that a residual block (He et al., 2016) can be interpreted as the discrete approximation of an ODE by setting the discretization step to be one. When the discretization step approaches zero, it yields a family of neural networks, which are called neural ODEs (Chen et al., 2018). Formally, in a neural ODE, the relation between input and output is characterized by the following set of equations:

$$\frac{\mathrm{d}\mathbf{z}(t)}{\mathrm{d}t} = f_\theta(\mathbf{z}(t), t), \quad \mathbf{z}(0) = \mathbf{z}_{\text{in}}, \quad \mathbf{z}_{\text{out}} = \mathbf{z}(T), \tag{1}$$

where $f_\theta : \mathbb{R}^d \times [0, \infty) \rightarrow \mathbb{R}^d$ denotes the trainable layers that are parameterized by weights $\theta$ and $\mathbf{z} : [0, \infty) \rightarrow \mathbb{R}^d$ represents the $d$-dimensional state of the neural ODE. We assume that $f_\theta$ is continuous in $t$ and globally Lipschitz continuous in $\mathbf{z}$. In this case, the input $\mathbf{z}_{\text{in}}$ of the neural ODE corresponds to the state at $t = 0$, and the output $\mathbf{z}_{\text{out}}$ is associated to the state at some $T \in (0, \infty)$. Because $f_\theta$ governs how the state changes with respect to time $t$, we also use $f_\theta$ to denote the *dynamics* of the neural ODE.

Given input $\mathbf{z}_{\text{in}}$, the output $\mathbf{z}_{\text{out}}$ can be computed by solving the ODE in (1). If $T$ is fixed, the output $\mathbf{z}_{\text{out}}$ only depends on the input $\mathbf{z}_{\text{in}}$ and the dynamics $f_\theta$, which also corresponds to the weighted layers in the neural ODE. Therefore, the neural ODE can be represented as the $d$-dimensional function $\phi_T(\cdot, \cdot)$ of the input $\mathbf{z}_{\text{in}}$ and the dynamics $f_\theta$, i.e.,

$$\mathbf{z}_{\text{out}} = \mathbf{z}(T) = \mathbf{z}(0) + \int_0^T f_\theta(\mathbf{z}(t), t) \, \mathrm{d}t = \phi_T(\mathbf{z}_{\text{in}}, f_\theta).$$

The terminal time $T$ of the output state $\mathbf{z}(T)$ is set to be 1 in practice. Several methods have been proposed for training neural ODEs, such as the adjoint sensitivity method (Chen et al., 2018), SNet (Quaglino et al., 2019), and the auto-differentiation technique (Paszke et al., 2017). In this work, we use the most straightforward technique, i.e., updating the weights $\theta$ with the auto-differentiation technique in the PyTorch framework.

## 3 AN EMPIRICAL STUDY ON THE ROBUSTNESS OF ODENETS

Robustness of deep models has gained increased attention, as it is imperative that deep models employed in critical applications, such as healthcare, are robust. The robustness of a model is measured by the sensitivity of the prediction with respect to small perturbations on the inputs. In this study, we consider three commonly-used perturbation schemes, namely random Gaussian perturbations, FGSM (Goodfellow et al., 2014) adversarial examples, and PGD (Madry et al., 2017) adversarial examples. These perturbation schemes reflect noise and adversarial robustness properties of the investigated models respectively. We evaluate the robustness via the classification accuracies on perturbed images, in which the original non-perturbed versions of these images are all correctly classified.

For a fair comparison with conventional CNN models, we made sure that the number of parameters of an ODENet is close to that of its counterpart CNN model. Specifically, the ODENet shares the same network architecture with the CNN model for the FE and FCC parts. The only difference is that, for the RM part, the input of the ODE-based RM is concatenated with one more channel which represents the time $t$, while the RM in a CNN model has a skip connection and serves as a residual block. During the training phase, all the hyperparameters are kept the same, including training epochs, learning rate schedules, and weight decay coefficients. Each model is trained *three* times with different random seeds, and we report the average performance (classification accuracy) together with the standard deviation.

### 3.1 EXPERIMENTAL SETTINGS

**Dataset:** We conduct experiments to compare the robustness of ODENets with CNN models on three datasets, i.e., the MNIST (LeCun et al., 1998), the SVHN (Netzer et al., 2011), and a subset of the ImageNet datset (Deng et al., 2009). We call the subset ImgNet10 since it is collected from 10 synsets of ImageNet: dog, bird, car, fish, monkey, turtle, lizard, bridge, cow, and crab. We selected 3,000 training images and 300 test images from each synset and resized all images to $128 \times 128$.

**Architectures:** On the MNIST dataset, both the ODENet and the CNN model consists of four convolutional layers and one fully-connected layer. The total number of parameters of the two models is around 140k. On the SVHN dataset, the networks are similar to those for the MNIST; we only changed the input channels of the first convolutional layer to three. On the ImgNet10 dataset, there are nine convolutional layers and one fully-connected layer for both the ODENet and the CNN model. The numbers of parameters is approximately 280k. In practice, the neural ODE can be solved with different numerical solvers such as the Euler method and the Runge-Kutta methods (Chen et al., 2018). Here, we use the easily-implemented Euler method in the experiments. To balance the computation and the continuity of the flow, we solve the ODE initial value problem in equation (1) by the Euler method with step size 0.1. Our implementation builds on the open-source neural ODE codes.[1] Details on the network architectures are included in the Appendix.

**Training:** The experiments are conducted using two settings on each dataset—training models only with original non-perturbed images and training models on original images together with their perturbed versions. In both settings, we added a weight decay term into the training objective to regularize the norm of the weights, since this can help control the model's representation capacity and improve the robustness of a neural network (Sokolić et al., 2017). In the second setting, images perturbed with random Gaussian noise are used to fine-tune the models, because augmenting the dataset with small perturbations can possibly improve the robustness of models and synthesizing Gaussian noise does not incur excessive computation time.

## 3.2 ROBUSTNESS OF ODENETS TRAINED ONLY ON NON-PERTURBED IMAGES

The first question we are interested in is how robust ODENets are against perturbations if the model is only trained on original non-perturbed images. We train CNNs and ODEnets to perform classification on three datasets and set the weight decay parameters for all models to be 0.0005. We make sure that both the well-trained ODENets and CNN models have satisfactory performances on original non-perturbed images, i.e., around 99.5% for MNIST, 95.0% for the SVHN, and 80.0% for ImgNet10.

Since Gaussian noise is ubiquitous in modeling image degradation, we first evaluated the robustness of the models in the presence of zero-mean random Gaussian perturbations. It has also been shown that a deep model is vulnerable to harmful adversarial examples, such as the FGSM (Goodfellow et al., 2014). We are also interested in how robust ODENets are in the presence of adversarial examples. The standard deviation $\sigma$ of Gaussian noise and the $l_\infty$-norm $\epsilon$ of the FGSM attack for each dataset are shown in Table 1.

Table 1: Robustness comparison of different models. We report their mean classification accuracies (%) and standard deviations (mean $\pm$ std) on perturbed images from the MNIST, the SVHN, and the ImgNet10 datasets. Two types of perturbations are used—zero-mean Gaussian noise and FGSM adversarial attack. The results show that ODENets are much more robust in comparison to CNN models.

| | Gaussian noise | | | Adversarial attack | | |
|---|---|---|---|---|---|---|
| MNIST | $\sigma = 50$ | $\sigma = 75$ | $\sigma = 100$ | FGSM-0.15 | FGSM-0.3 | FGSM-0.5 |
| CNN | 98.1±0.7 | 85.8±4.3 | 56.4±5.6 | 63.4±2.3 | 24.0±8.9 | 8.3±3.2 |
| ODENet | **98.7**±0.6 | **90.6**±5.4 | **73.2**±8.6 | **83.5**±0.9 | **42.1**±2.4 | **14.3**±2.1 |
| SVHN | $\sigma = 15$ | $\sigma = 25$ | $\sigma = 35$ | FGSM-3/255 | FGSM-5/255 | FGSM-8/255 |
| CNN | 90.0±1.2 | 76.3±2.7 | 60.9±3.9 | 29.2±2.9 | 13.7±1.9 | 5.4±1.5 |
| ODENet | **95.7**±0.7 | **88.1**±1.5 | **78.2**±2.1 | **58.2**±2.3 | **43.0**±1.3 | **30.9**±1.4 |
| ImgNet10 | $\sigma = 10$ | $\sigma = 15$ | $\sigma = 25$ | FGSM-5/255 | FGSM-8/255 | FGSM-16/255 |
| CNN | 80.1±1.8 | 63.3±2.0 | 40.8±2.7 | 28.5±0.5 | 18.1±0.7 | 9.4±1.2 |
| ODENet | **81.9**±2.0 | **67.5**±2.0 | **48.7**±2.6 | **36.2**±1.0 | **27.2**±1.1 | **14.4**±1.7 |

From the results in Table 1, we observe that the ODENets demonstrate superior robustness compared to CNNs for all types of perturbations. On the MNIST dataset, in the presence of Gaussian

---

[1] https://github.com/rtqichen/torchdiffeq.

perturbations with a large $\sigma$ of 100, the ODENet produces much higher accuracy on perturbed images compared to the CNN model (73.2% vs. 56.4%). For the FGSM-0.3 adversarial examples, the accuracy of ONEnet is around twice as high as that of the CNN model. On the SVHN dataset, ODENets significantly outperform CNN models, e.g., for the FGSM-5/255 examples, the accuracy of the ODENet is 43.0%, which is much higher than that of the CNN model (13.7%). On the ImgNet10, for both cases of $\sigma = 25$ and FGSM-8/255, ODENet outperforms CNNs by a large margin of around 9%.

## 3.3 ROBUSTNESS OF ODENETS TRAINED ON ORIGINAL IMAGES TOGETHER WITH GAUSSIAN PERTURBATIONS

Training a model on original images together with their perturbed versions can improve the robustness of the model. As mentioned previously, Gaussian noise is commonly assumed to be present in real-world images. Synthesizing Gaussian noise is also fast and easy. Thus, we add random Gaussian noise into the original images to generate their perturbed versions. ODENets and CNN models are both trained on original images together with their perturbed versions. The standard deviation of the added Gaussian noise is randomly chosen from $\{50, 75, 100\}$ on the MNIST dataset, $\{15, 25, 35\}$ on the SVHN dataset, and $\{10, 15, 25\}$ on the ImgNet10. All other hyperparameters are kept the same as above.

Table 2: Robustness comparison of different models. We report their mean classification accuracies (%) and standard deviations (mean $\pm$ std) on perturbed images from the MNIST, the SVHN, and the ImgNet10 datsets. Three types of perturbations are used—zero-mean Gaussian noise, FGSM adversarial attack, and PGD adversarial attack. The results show that ODENets are more robust compared to CNN models.

| | Gaussian noise | Adversarial attack | | | |
|---|---|---|---|---|---|
| MNIST | $\sigma = 100$ | FGSM-0.3 | FGSM-0.5 | PGD-0.2 | PGD-0.3 |
| CNN | 98.7±0.1 | 54.2±1.1 | 15.8±1.3 | 32.9±3.7 | 0.0±0.0 |
| ODENet | **99.4**±0.1 | **71.5**±1.1 | **19.9**±1.2 | **64.7**±1.8 | **13.0**±0.2 |
| SVHN | $\sigma = 35$ | FGSM-5/255 | FGSM-8/255 | PGD-3/255 | PGD-5/255 |
| CNN | 90.6±0.2 | 25.3±0.6 | 12.3±0.7 | 32.4±0.4 | 14.0±0.5 |
| ODENet | **95.1**±0.1 | **49.4**±1.0 | **34.7**±0.5 | **50.9**±1.3 | **27.2**±1.4 |
| ImgNet10 | $\sigma = 25$ | FGSM-5/255 | FGSM-8/255 | PGD-3/255 | PGD-5/255 |
| CNN | 92.6±0.6 | 40.9±1.8 | 26.7±1.7 | 28.6±1.5 | 11.2±1.2 |
| ODENet | 92.6±0.5 | **42.0**±0.4 | **29.0**±1.0 | **29.8**±0.4 | **12.3**±0.6 |

The robustness of the models is evaluated under Gaussian perturbations, FGSM adversarial examples, and PGD (Madry et al., 2017) adversarial examples. The latter is a stronger attacker compared to the FGSM. The $l_\infty$-norm $\epsilon$ of the PGD attack for each dataset is shown in Table 2. Based on the results, we observe that ODENets consistently outperform CNN models on both two datasets. On the MNIST dataset, the ODENet outperforms the CNN against all types of perturbations. In particular, for the PGD-0.2 adversarial examples, the accuracy of the ODENet (64.7%) is much higher than that of the CNN (32.9%). Besides, for the PGD-0.3 attack, the CNN is completely misled by the adversarial examples, but the ODENet can still classify perturbed images with an accuracy of 13.0%. On the SVHN dataset, ODENets also show superior robustness in comparison to CNN models. For all the adversarial examples, ODENets outperform CNN models by a margin of at least 10 percentage points. On the ImgNet10 dataset, the ODENet also performs better than CNN models against all forms of adversarial examples.

## 3.4 INSIGHTS ON THE ROBUSTNESS OF ODENETS

From the results in Sections 3.2 and 3.3, we find ODENets are more robust compared to CNN models. Here, we attempt to provide an intuitive understanding of the robustness of the neural ODE. In an ODENet, given some datum, the FE extracts an informative feature map from the datum. The neural ODE, serving as the RM, takes as input the feature map and performs a nonlinear mapping. In practice, we use the weight decay technique during training which regularizes the norm of weights

in the FE part, so that the change of feature map in terms of a small perturbation on the input can be controlled. We aim to show that, in the neural ODE, a small change on the feature map will not lead to a large deviation from the original output associated with the feature map.

**Theorem 1** (ODE integral curves do not intersect (Coddington & Levinson, 1955; Younes, 2010; Dupont et al., 2019)). *Let $\mathbf{z}_1(t)$ and $\mathbf{z}_2(t)$ be two solutions of the ODE in (1) with different initial conditions, i.e. $\mathbf{z}_1(0) \neq \mathbf{z}_2(0)$. In (1), $f_\theta$ is continuous in $t$ and globally Lipschitz continuous in $\mathbf{z}$. Then, it holds that $\mathbf{z}_1(t) \neq \mathbf{z}_2(t)$ for all $t \in [0, \infty)$.*

To illustrate this theorem, considering a simple 1-dimensional system in which the state is a scalar. As shown in Figure 2, equation (1) has a solution $z_1(t)$ starting from $A_1 = (0, z_1(0))$, where $z_1(0)$ is the feature of some datum. Equation (1) also has another two solutions $z_2(t)$ and $z_3(t)$, whose starting points $A_2 = (0, z_2(0))$ and $A_3 = (0, z_3(0))$, both of which are close to $A_1$. Suppose $A_1$ is between $A_2$ and $A_3$. By Theorem 1, we know that the integral curve $z_1(t)$ is always sandwiched between the integral curves $z_2(t)$ and $z_3(t)$.

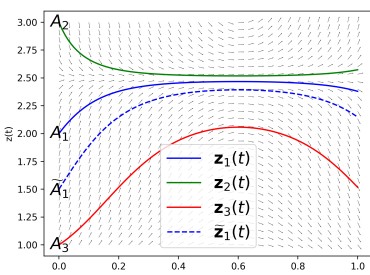

Figure 2: No integral curves intersect. The integral curve starting from $\widetilde{A}_1$ is always sandwiched between two integral curves starting from $A_1$ and $A_3$.

Now, let $\epsilon < \min\{|z_2(0) - z_1(0)|, |z_3(0) - z_1(0)|\}$. Consider a solution $\widetilde{z}_1(t)$ of equation (1). The integral curve $\widetilde{z}_1(t)$ starts from a point $\widetilde{A}_1 = (0, \widetilde{z}_1(0))$. The point $\widetilde{A}_1$ is in the $\epsilon$-neighborhood of $A_1$ with $|\widetilde{z}_1(0) - z_1(0)| < \epsilon$. By Theorem 1, we know that $|\widetilde{z}_1(T) - z_1(T)| \leq |z_3(T) - z_2(T)|$. In other words, if any perturbation smaller than $\epsilon$ is added to the scalar $z_1(0)$ in $A_1$, the deviation from the original output $z_1(T)$ is bounded by the distance between $z_2(T)$ and $z_3(T)$. In contrast, in a CNN model, there is no such bound on the deviation from the original output. Thus, we opine that due to this non-intersecting property, ODENets are intrinsically robust.

## 4 TISODE: BOOSTING THE ROBUSTNESS OF NEURAL ODES

In the previous section, we presented an empirical study on the robustness of ODENets and observed that ODENets are more robust compared to CNN models. In this section, we explore how to boost the robustness of the vanilla neural ODE model further. This motivates the proposal of *time-invariant steady neural ODEs* (TisODEs).

### 4.1 TIME-INVARIANT STEADY NEURAL ODES

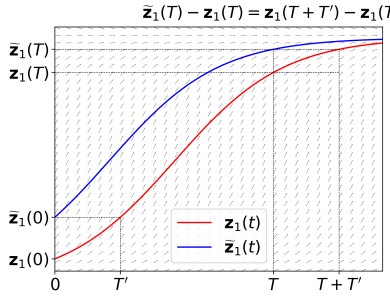

Figure 3: An illustration of the time-invariant property of ODEs. We can see that the curve $\widetilde{\mathbf{z}}_1(t)$ is exactly the horizontal translation of $\mathbf{z}_1(t)$ on the interval $[T', \infty)$.

From the discussion in Section 3.4, the key to improving the robustness of neural ODEs is to control the difference between neighboring integral curves. By Grownall's inequality (Howard, 1998) (see Theorem 2 in the Appendix), we know that the difference between two terminal states is bounded by the difference between initial states multiplied by the exponential of the dynamics' Lipschitz constant. However, it is very difficult to bound the Lipschitz constant of the dynamics directly. Alternatively, we propose to achieve the goal of controlling the output deviation by following two steps: (i) removing the time dependence of the dynamics and (ii) imposing a certain steady-state constraint.

In the neural ODE characterized by equation (1), the dynamics $f_\theta(\mathbf{z}(t), t)$ depends on both the state $\mathbf{z}(t)$ at time $t$ and the time $t$ itself. In contrast, if the neural ODE is modified to be time-invariant, the time dependence of the dynamics is removed. Consequently, the dynamics depends *only* on the state $\mathbf{z}$. So, we can rewrite

the dynamics function as $f_\theta(\mathbf{z})$, and the neural ODE is characterized as

$$\begin{cases} \dfrac{d\mathbf{z}(t)}{dt} = f_\theta(\mathbf{z}(t)); \\ \mathbf{z}(0) = \mathbf{z}_{\text{in}}; \\ \mathbf{z}_{\text{out}} = \mathbf{z}(T). \end{cases} \tag{2}$$

Let $\mathbf{z}_1(t)$ be a solution of (2) on $[0, \infty)$ and $\epsilon > 0$ be a small positive value. We define the set $\mathbb{M}_1 = \{(\mathbf{z}_1(t), t) | t \in [0, T], \|\mathbf{z}_1(t) - \mathbf{z}_1(0)\| \le \epsilon\}$. This set contains all points on the curve of $\mathbf{z}_1(t)$ during $[0, T]$ that are also inside the $\epsilon$-neighborhood of $\mathbf{z}_1(0)$. For some element $(\mathbf{z}_1(T'), T') \in \mathbb{M}_1$, let $\widetilde{\mathbf{z}}_1(t)$ be the solution of (2) which starts from $\widetilde{\mathbf{z}}_1(0) = \mathbf{z}_1(T')$. Then we have

$$\widetilde{\mathbf{z}}_1(t) = \mathbf{z}_1(t + T') \tag{3}$$

for all $t$ in $[0, \infty)$. The property shown in equation (3) is known as the *time-invariant property*. It indicates that the integral curve $\widetilde{\mathbf{z}}_1(t)$ is the $-T'$ shift of $\mathbf{z}_1(t)$ (Figure 3).

We can regard $\widetilde{\mathbf{z}}_1(0)$ as a slightly perturbed version of $\mathbf{z}_1(0)$, and we are interested in how large the difference between $\widetilde{\mathbf{z}}_1(T)$ and $\mathbf{z}_1(T)$ is. In a robust model, the difference should be small. By equation (3), we have $\|\widetilde{\mathbf{z}}_1(T) - \mathbf{z}_1(T)\| = \|\mathbf{z}_1(T + T') - \mathbf{z}_1(T)\|$. Since $T' \in [0, T]$, the difference between $\mathbf{z}_1(T)$ and $\widetilde{\mathbf{z}}_1(T)$ can be bounded as follows,

$$\|\widetilde{\mathbf{z}}_1(T) - \mathbf{z}_1(T)\| = \left\| \int_T^{T+T'} f_\theta(\mathbf{z}_1(t)) \, dt \right\| \le \left\| \int_T^{T+T'} |f_\theta(\mathbf{z}_1(t))| \, dt \right\| \le \left\| \int_T^{2T} |f_\theta(\mathbf{z}_1(t))| \, dt \right\|, \tag{4}$$

where all norms are $\ell_2$ norms and $|f_\theta|$ denotes the element-wise absolute operation of a vector-valued function $f_\theta$. That is to say, the difference between $\widetilde{\mathbf{z}}_1(T)$ and $\mathbf{z}_1(T)$ can be bounded by only using the information of the curve $\mathbf{z}_1(t)$. For any $t' \in [0, T]$ and element $(\mathbf{z}_1(t'), t') \in \mathbb{M}_1$, consider the integral curve that starts from $\mathbf{z}_1(t')$. The difference between the output state of this curve and $\mathbf{z}_1(T)$ satisfies inequality (4).

Therefore, we propose to add an additional term $L_{\text{ss}}$ to the loss function when training the time-invariant neural ODE:

$$L_{\text{ss}} = \sum_{i=1}^{N} \left\| \int_T^{2T} |f_\theta(\mathbf{z}_i(t))| \, dt \right\|, \tag{5}$$

where $N$ is the number of samples in the training set and $\mathbf{z}_i(t)$ is the solution whose initial state equals to the feature of the $i^{\text{th}}$ sample. The regularization term $L_{\text{ss}}$ is termed as the steady-state loss. This terminology "steady state" is borrowed from the dynamical systems literature. In a stable dynamical system, the states stabilize around a fixed point, known as the *steady-state*, as time tends to infinity. If we can ensure that $L_{\text{ss}}$ is small, for each sample, the outputs of all the points in $\mathbb{M}_i$ will stabilize around $\mathbf{z}_i(T)$. Consequently, the model is robust. This modification of the neural ODE is dubbed *Time-invariant steady neural ODE*.

## 4.2 EVALUATING ROBUSTNESS OF TISODE-BASED CLASSIFIERS

Here, we conduct experiments to evaluate the robustness of our proposed TisODE, and compare TisODE-based models with the vanilla ODENets. We train all models with original non-perturbed images together with their Gaussian perturbed versions. The regularization parameter for the steady-state loss $L_{\text{ss}}$ is set to be $0.1$. All other hyperparameters are exactly the same as those in Section 3.3.

From the results in Table 3, we can see that our proposed TisODE-based models are clearly more robust compared to vanilla ODENets. On the MNIST dataset, when combating FGSM-0.3 attacks, the TisODE-based models outperform vanilla ODENets by more than 4 percentage points. For the FGSM-0.5 adversarial examples, the accuracy of the TisODE-based model is 6 percentage points better. On the SVHN dataset, the TisODE-based models perform better in terms of all forms of adversarial examples. On the ImgNet10 dataset, the TisODE-based models also outperform vanilla ODE-based models on all types of perturbations. In the presence of FGSM and PGD-5/255 examples, the accuracies are enhanced by more than 2 percentage points.

Table 3: Classification accuracy (mean ± std in %) on perturbed images from MNIST, SVHN and ImgNet10. To evaluate the robustness of classifiers, we use three types of perturbations, namely zero-mean Gaussian noise with standard deviation $\sigma$, FGSM attack and PGD attack. From the results, the proposed TisODE effectively improve the robustness of the vanilla neural ODE.

| | Gaussian noise | Adversarial attack | | | |
|---|---|---|---|---|---|
| MNIST | $\sigma = 100$ | FGSM-0.3 | FGSM-0.5 | PGD-0.2 | PGD-0.3 |
| CNN | 98.7±0.1 | 54.2±1.1 | 15.8±1.3 | 32.9±3.7 | 0.0±0.0 |
| ODENet | 99.4±0.1 | 71.5±1.1 | 19.9±1.2 | 64.7±1.8 | 13.0±0.2 |
| TisODE | **99.6**±0.0 | **75.7**±1.4 | **26.5**±3.8 | **67.4**±1.5 | **13.2**±1.0 |
| SVHN | $\sigma = 35$ | FGSM-5/255 | FGSM-8/255 | PGD-3/255 | PGD-5/255 |
| CNN | 90.6±0.2 | 25.3±0.6 | 12.3±0.7 | 32.4±0.4 | 14.0±0.5 |
| ODENet | **95.1**±0.1 | 49.4±1.0 | 34.7±0.5 | 50.9±1.3 | 27.2±1.4 |
| TisODE | 94.9±0.1 | **51.6**±1.2 | **38.2**±1.9 | **52.0**±0.9 | **28.2**±0.3 |
| ImgNet10 | $\sigma = 25$ | FGSM-5/255 | FGSM-8/255 | PGD-3/255 | PGD-5/255 |
| CNN | 92.6±0.6 | 40.9±1.8 | 26.7±1.7 | 28.6±1.5 | 11.2±1.2 |
| ODENet | 92.6±0.5 | 42.0±0.4 | 29.0±1.0 | 29.8±0.4 | 12.3±0.6 |
| TisODE | **92.8**±0.4 | **44.3**±0.7 | **31.4**±1.1 | **31.1**±1.2 | **14.5**±1.1 |

### 4.3 TISODE - A GENERALLY APPLICABLE DROP-IN TECHNIQUE FOR IMPROVING THE ROBUSTNESS OF DEEP NETWORKS

In view of the excellent robustness of the TisODE, we claim that the proposed TisODE can be used as a general drop-in module for improving the robustness of deep networks. We support this claim by showing the TisODE can work in conjunction with other state-of-the-art techniques and further boost the models' robustness. These techniques include the feature denoising (FDn) method (Xie et al., 2019) and the input randomization (IR) method (Xie et al., 2017). We conduct experiments on the MNIST and SVHN datasets. All models are trained with original non-perturbed images together with their Gaussian perturbed versions. We show that models using the FDn/IRd technique becomes much more robust when equipped with the TisODE. In the FDn experiments, the dot-product non-local denoising layer (Xie et al., 2019) is added to the head of the fully-connected classifier.

Table 4: Classification accuracy (mean ± std in %) on perturbed images from MNIST and SVHN. We evaluate against three types of perturbations, namely zero-mean Gaussian noise with standard deviation $\sigma$, FGSM attack and PGD attack. From the results, upon the CNNs modified with FDn and IRd, using TisODE can further improve the robustness.

| | Gaussian noise | Adversarial attack | | | |
|---|---|---|---|---|---|
| MNIST | $\sigma = 100$ | FGSM-0.3 | FGSM-0.5 | PGD-0.2 | PGD-0.3 |
| CNN | 98.7±0.1 | 54.2±1.1 | 15.8±1.3 | 32.9±3.7 | 0.0±0.0 |
| CNN-FDn | 99.0±0.1 | 74.0±4.1 | 32.6±5.3 | 58.9±4.0 | 8.2±2.6 |
| TisODE-FDn | **99.4**±0.0 | 80.6±2.3 | 40.4±5.7 | 72.6±2.4 | 28.2±3.6 |
| CNN-IRd | 95.3±0.9 | 78.1±2.2 | 36.7±2.1 | 79.6±1.9 | 55.5±2.9 |
| TisODE-IRd | 97.6±0.1 | **86.8**±2.3 | **49.1**±0.2 | **88.8**±0.9 | **66.0**±0.9 |
| SVHN | $\sigma = 35$ | FGSM-5/255 | FGSM-8/255 | PGD-3/255 | PGD-5/255 |
| CNN | 90.6±0.2 | 25.3±0.6 | 12.3±0.7 | 32.4±0.4 | 14.0±0.5 |
| CNN-FDn | 92.4±0.1 | 43.8±1.4 | 31.5±3.0 | 40.0±2.6 | 19.6±3.4 |
| TisODE-FDn | **95.2**±0.1 | 57.8±1.7 | 48.2±2.0 | 53.4±2.9 | 32.3±1.0 |
| CNN-IRd | 84.9±1.2 | 65.8±0.4 | 54.7±1.2 | 74.0±0.5 | 64.5±0.8 |
| TisODE-IRd | 91.7±0.5 | **74.4**±1.2 | **61.9**±1.8 | **81.6**±0.8 | **71.0**±0.5 |

From Table 4, we observe that both FDn and IRd can effectively improve the adversarial robustness of vanilla CNN models (CNN-FDn, CNN-IRd). Furthermore, combining our proposed TisODE with FDn or IRd (TisODE-FDn, TisODE-IRd), the adversarial robustness of the resultant model is significantly enhanced. For example, on the MNIST dataset, the additional use of our TisODE increases the accuracies on the PGD-0.3 examples by at least 10 percentage points for both FDn (8.2% to 28.2%) and IRd (55.5% to 66.0%). However, on both MNIST and SVHN datasets, the IRd technique improves the robustness against adversarial examples, but its performance is worse

on random Gaussian noise. With the help of the TisODE, the degradation in the robustness against random Gaussian noise can be effectively ameliorated.

## 5 RELATED WORKS

In this section, we briefly review related works on the neural ODE and works concerning improving the robustness of deep neural networks.

**Neural ODE:** The neural ODE (Chen et al., 2018) method models the input and output as two states of a continuous-time dynamical system by approximating the dynamics of this system with trainable layers. Before the proposal of neural ODE, the idea of modeling nonlinear mappings using continuous-time dynamical systems was proposed in Weinan (2017). Lu et al. (2017) also showed that several popular network architectures could be interpreted as the discretization of a continuous-time ODE. For example, the ResNet (He et al., 2016) and PolyNet (Zhang et al., 2017) are associated with the Euler scheme and the FractalNet (Larsson et al., 2016) is related to the Runge-Kutta scheme. In contrast to these discretization models, neural ODEs are endowed with an intrinsic invertibility property, which yields a family of invertible models for solving inverse problems (Ardizzone et al., 2018), such as the FFJORD (Grathwohl et al., 2018).

Recently, many researchers have conducted studies on neural ODEs from the perspectives of optimization techniques, approximation capabilities, and generalization. Concerning the optimization of neural ODEs, the auto-differentiation techniques can effectively train ODENets, but the training procedure is computationally and memory inefficient. To address this problem, Chen et al. (2018) proposed to compute gradients using the adjoint sensitivity method (Pontryagin, 2018), in which there is no need to store any intermediate quantities of the forward pass. Also in Quaglino et al. (2019), the authors proposed the SNet which accelerates the neural ODEs by expressing their dynamics as truncated series of Legendre polynomials. Concerning the approximation capability, Dupont et al. (2019) pointed out the limitations in approximation capabilities of neural ODEs because of the preserving of input topology. The authors proposed an augmented neural ODE which increases the dimension of states by concatenating zeros so that complex mappings can be learned with simple flow. The most relevant work to ours concerns strategies to improve the generalization of neural ODEs. In Liu et al. (2019), the authors proposed the neural stochastic differential equation (SDE) by injecting random noise to the dynamics function and showed that the generalization and robustness of vanilla neural ODEs could be improved. However, our improvement on the neural ODEs is explored from a different perspective by introducing constraints on the flow. We empirically found that our proposal and the neural SDE can work in tandem to further boost the robustness of neural ODEs.

**Robust Improvement:** A straightforward way of improving the robustness of a model is to smooth the loss surface by controlling the spectral norm of the Jacobian matrix of the loss function (Sokolić et al., 2017). In terms of adversarial examples (Carlini & Wagner, 2017; Chen et al., 2017), researchers have proposed adversarial training strategies (Madry et al., 2017; Elsayed et al., 2018; Tramèr et al., 2017) in which the model is fine-tuned with adversarial examples generated in real-time. However, generating adversarial examples is not computationally efficient, and there exists a trade-off between the adversarial robustness and the performance on original non-perturbed images (Yan et al., 2018; Tsipras et al., 2018). In Wang et al. (2018a), the authors model the ResNet as a transport equation, in which the adversarial vulnerability can be interpreted as the irregularity of the decision boundary. Consequently, a diffusion term is introduced to enhance the robustness of the neural nets. Besides, there are also some works that propose novel architectural defense mechanisms against adversarial examples. For example, Xie et al. (2017) utilized random resizing and random padding to destroy the specific structure of adversarial perturbations; Wang et al. (2018b) and Wang et al. (2018c) improved the robustness of neural networks by replacing the output layers with novel interpolating functions; In Xie et al. (2019), the authors designed a feature denoising filter that can remove the perturbation's pattern from feature maps. In this work, we explore the intrinsic robustness of a specific novel architecture (neural ODE), and show that the proposed TisODE can improve the robustness of deep networks and can also work in tandem with these state-of-the-art methods Xie et al. (2017; 2019) to achieve further improvements.

## 6 CONCLUSION

In this paper, we first empirically study the robustness of neural ODEs. Our studies reveal that neural ODE-based models are superior in terms of robustness compared to CNN models. We then explore how to further boost the robustness of vanilla neural ODEs and propose the TisODE. Finally, we show that the proposed TisODE outperforms the vanilla neural ODE and also can work in conjunction with other state-of-the-art techniques to further improve the robustness of deep networks. Thus, the TisODE method is an effective drop-in module for building robust deep models.

## ACKNOWLEDGEMENT

This work is funded by a Singapore National Research Foundation (NRF) Fellowship (R-263-000-D02-281).

Jiashi Feng was partially supported by NUS IDS R-263-000-C67-646, ECRA R-263-000-C87-133, MOE Tier-II R-263-000-D17-112 and AI.SG R-263-000-D97-490

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

# 7 APPENDIX

## 7.1 NETWORKS USED ON THE MNIST, THE SVHN, AND THE IMGNET10 DATASETS

Table 5: The architectures of the ODENets on different datasets.

| MNIST | Repetition | Layer |
|---|---|---|
| FE | ×1 | Conv(1, 64, 3, 1) + GroupNorm + ReLU |
|  | ×1 | Conv(64, 64, 4, 2) + GroupNorm + ReLU |
| RM | ×2 | Conv(64, 64, 3, 1) + GroupNorm + ReLU |
| FCC | ×1 | AdaptiveAvgPool2d + Linear(64,10) |

| SVHN | Repetition | Layer |
|---|---|---|
| FE | ×1 | Conv(3, 64, 3, 1) + GroupNorm + ReLU |
|  | ×1 | Conv(64, 64, 4, 2) + GroupNorm + ReLU |
| RM | ×2 | Conv(64, 64, 3, 1) + GroupNorm + ReLU |
| FCC | ×1 | AdaptiveAvgPool2d + Linear(64,10) |

| ImgNet10 | Repetition | Layer |
|---|---|---|
| FE | ×1 | Conv(3, 32, 5, 2) + GroupNorm |
|  | ×1 | MaxPooling(2) |
|  | ×1 | BaiscBlock(32, 64, 2) |
|  | ×1 | MaxPooling(2) |
| RM | ×3 | BaiscBlock(64, 64, 1) |
| FCC | ×1 | AdaptiveAvgPool2d + Linear(64,10) |

In Table 5, the four arguments of the Conv layer represent the input channel, output channel, kernel size, and the stride. The two arguments of the Linear layer represents the input dimension and the output dimension of this fully-connected layer. In the network on the ImgNet10, the BasicBlock refers to the standard architecture in (He et al., 2016), the three arguments of the BasicBlock represent the input channel, output channel and the stride of the Conv layers inside the block. Note that we replace the BatchNorm layers in BasicBlocks as the GroupNorm to guarantee that the dynamics of each datum is independent of other data in the same mini-batch.

## 7.2 THE CONSTRUCTION OF IMGNET10 DATASET

Table 6: The corresponding indexes to each class in the original ImageNet dataset

| Class | Indexing | | |
|---|---|---|---|
| dog | n02090721, | n02091032, | n02088094 |
| bird | n01532829, | n01558993, | n01534433 |
| car | n02814533, | n03930630, | n03100240 |
| fish | n01484850, | n01491361, | n01494475 |
| monkey | n02483708, | n02484975, | n02486261 |
| turtle | n01664065, | n01665541, | n01667114 |
| lizard | n01677366, | n01682714, | n01685808 |
| bridge | n03933933, | n04366367, | n04311004 |
| cow | n02403003, | n02408429, | n02410509 |
| crab | n01980166, | n01978455, | n01981276 |

## 7.3 GRONWALL'S INEQUALITY

We formally state the Gronwall's Inequality here, following the version in (Howard, 1998).

**Theorem 2.** *Let $U \subset \mathbb{R}^d$ be an open set. Let $f : U \times [0, T] \to \mathbb{R}^d$ be a continuous function and let $\mathbf{z}_1, \mathbf{z}_2: [0, T] \to U$ satisfy the initial value problems:*

$$\frac{d\mathbf{z}_1(t)}{dt} = f(\mathbf{z}_1(t), t), \quad \mathbf{z}_1(t) = \mathbf{x}_1$$

$$\frac{d\mathbf{z}_2(t)}{dt} = f(\mathbf{z}_2(t), t), \quad \mathbf{z}_2(t) = \mathbf{x}_2$$

*Assume there is a constant $C \geq 0$ such that, for all $t \in [0, T]$,*

$$\|f(\mathbf{z}_2(t), t) - f(\mathbf{z}_1(t), t))\| \leq C\|\mathbf{z}_2(t) - \mathbf{z}_1(t)\|$$

*Then, for any $t \in [0, T]$,*

$$\|\mathbf{z}_1(t) - \mathbf{z}_2(t)\| \leq \|\mathbf{x}_2 - \mathbf{x}_1\| \cdot e^{Ct}.$$

## 7.4 MORE EXPERIMENTAL RESULTS

### 7.4.1 COMPARISON IN THE SETTING OF ADVERSARIAL TRAINING

We implement the adversarial training of the models on the MNIST dataset, and the adversarial examples for training are generated in real-time via the FGSM method (epsilon=0.3) during each epoch (Madry et al., 2017). The results of the adversarially trained models are shown in Table 7. We can observe that the neural ODE-based models are consistently more robust than CNN models. The proposed TisODE also outperforms the vanilla neural ODE.

Table 7: Classification accuracy (%) on perturbed images from MNIST. To evaluate the robustness of classifiers, we use three types of perturbations, namely zero-mean Gaussian noise with standard deviation $\sigma$, FGSM attack and PGD attack.

|  | Gaussian noise | Adversarial attack | | |
|---|---|---|---|---|
| MNIST | $\sigma = 100$ | FGSM-0.3 | FGSM-0.5 | PGD-0.3 |
| CNN | 58.0 | 98.4 | 21.1 | 5.3 |
| ODENet | 84.2 | 99.1 | 36.0 | 12.3 |
| TisODE | **87.9** | **99.1** | **66.5** | **78.9** |

### 7.4.2 EXPERIMENTS ON THE CIFAR10 DATASET

We conduct experiments on CIFAR10 to compare the robustness of CNN and neural ODE-based models. We train all the models only with original non-perturbed images and evaluate the robustness of models against random Gaussian noise and FGSM adversarial attacks. The results are shown in Table 8. We can observe that the ONENet is more robust than the CNN model in terms of both the random noise and the FGSM attack. Besides, our proposal, TisODE, can improve the robustness of the vanilla neural ODE.

Table 8: Classification accuracy (%) on perturbed images from CIFAR10. To evaluate the robustness of classifiers, we use two types of perturbations, namely zero-mean Gaussian noise with standard deviation $\sigma$ and FGSM attack.

|  | Gaussian noise | | Adversarial attack | |
|---|---|---|---|---|
| CIFAR10 | $\sigma = 15$ | $\sigma = 20$ | FGSM-8/255 | FGSM-10/255 |
| CNN | 70.2 | 57.6 | 24.3 | 18.4 |
| ODENet | 72.6 | 60.6 | 31.2 | 26.0 |
| TisODE | **74.3** | **62.0** | **33.6** | **26.8** |

Here, we control the number of parameters to be the same for all kinds of models. We use a small network, which consists of five convolutional layers and one linear layer.

Table 9: The architecture of the ODENet on CIFAR10.

|  | Repetition | Layer |
|---|---|---|
| FE | ×1 | Conv(3, 16, 3, 1) + GroupNorm + ReLU |
|  | ×1 | Conv(16, 32, 3, 2) + GroupNorm + ReLU |
|  | ×1 | Conv(32, 64, 3, 2) + GroupNorm + ReLU |
| RM | ×2 | Conv(64, 64, 3, 1) + GroupNorm + ReLU |
| FCC | ×1 | AdaptiveAvgPool2d + Linear(64,10) |

### 7.4.3 AN EXTENSION ON THE COMPARISON BETWEEN CNNS AND ODENETS

Here, we compare CNN and neural ODE-based models by controlling both the number of parameters and the number of function evaluations. We conduct experiments on the MNIST dataset, and all the models are trained only with original non-perturbed images.

For the neural ODE-based models, the time range is set from 0 to 1. We use the Euler method, and the step size is set to be 0.05. Thus the number of evaluations is $1/0.05 = 20$. For the CNN models (specifically ResNet), we repeatedly concatenate the residual block for 20 times, and these 20 blocks share the same weights. Our experiments show that, in this condition, the neural ODE-based models still outperform the CNN models (FGSM-0.15: 87.5% vs. 81.9%, FGSM-0.3: 53.4% vs. 49.7%, PGD-0.2: 11.8% vs. 4.8%).

