# OpenReview forum: "On Robustness of Neural Ordinary Differential Equations"
_ICLR.cc/2020/Conference — Accept (Spotlight)_

### Official Review · AnonReviewer1 · 2019-10-14
**Official Blind Review #1**

**Rating:** 6

**Review:**

This paper studied the robustness of neural ODE-based networks (ODENets) to various types of perturbations on the input images. The authors observed that ODENets are more robust to both Gaussian perturbation and adversarial attacks, which the authors explained as non-intersecting of the integral curves for different initial points. Moreover, the authors proposed the time-invariant steady neural ODE (TisODE) to enhance the robustness of ODENets. I list my concerns below:

1. To show the ODENet is more robust, the authors should bound the gap between the integral curves for different inputs. Non-intersecting of the integral curves does not guarantee the robustness.


2. The ODENet architecture showed in Figure~1 can be regarded as an augmented CNN. I think the identity map gives a good trade-off between robustness and generalization. To enhance robustness, one might design an expansion map, but this, in general, hurt the accuracy of the model.

3. Why do not perform experiments on the CIFAR10 benchmark dataset? I think it is very important to add these results.

4. To verify the robustness of ODENets and CNNs, the authors should also perform adversarial training besides training on the original and noisy images with Gaussian perturbation.

5. Theorem~1 is wrong. I suggest the authors check the conditions to make it valid. We can construct an ODE of the form (1) that blows up in finite-time, e.g., dx/dt = x^2.

6. Most importantly, the authors should match the number of function evaluations of neural ODE with the depth of the CNN, in addition to matching the number of parameters. Please perform such a comparison in rebuttal. (THIS IS THE MOST IMPORTANT RESULT I WANT TO SEE IN REBUTTAL)

7. The authors did not compare with existing work that tries to improve the robustness of neural nets from a differential equation viewpoint. The related works should be elaborated.


======================
I would like to point out a few related papers that lift the dimension of ODE to a transport equation and improve the neural nets' robustness from the lens of the transport equation's theory. Also, the author should compare their results with some reported results in 1, 3, 4.

1. Bao Wang, Binjie Yuan, Zuoqiang Shi, Stanley J. Osher. ResNets Ensemble via the Feynman-Kac Formalism to Improve Natural and Robust Accuracies, arXiv:1811.10745, NeurIPS, 2019

2. Bao Wang, Xiyang Luo, Zhen Li, Wei Zhu, Zuoqiang Shi, Stanley J. Osher. Deep Neural Nets with Interpolating Function as Output Activation, NeurIPS, 2018

3. Bao Wang, Alex T. Lin, Zuoqiang Shi, Wei Zhu, Penghang Yin, Andrea L. Bertozzi, Stanley J. Osher. Adversarial Defense via Data Dependent Activation Function and Total Variation Minimization, arXiv:1809.08516, 2018

4. B. Wang, S. Osher. Graph Interpolating Activation Improves Both Natural and Robust Accuracies in Data-Efficient Deep Learning, arXiv:1907.06800

=======================
Please address the previously mentioned concerns in rebuttal.

**Experience Assessment:**

I have published in this field for several years.

**Review Assessment: Checking Correctness Of Derivations And Theory:**

I carefully checked the derivations and theory.

**Review Assessment: Checking Correctness Of Experiments:**

I carefully checked the experiments.

**Review Assessment: Thoroughness In Paper Reading:**

I read the paper thoroughly.

---

> ### Author Response · Authors · 2019-11-13
> **Response (AnonReviewer1)  - Part 1**
>
> Thanks a lot for the review and the helpful advice. We have added experimental results into the revision (Appendix section 7.4). We group the questions into four topics and answer them as follows :
>
> [### First ###] For Q.2, on the architecture of ODENets, we follow the standard design in [5], which originally proposes the neural ODE method.

---

> > ### Comment · AnonReviewer1 · 2019-11-13
> > **Thank you for your reply**
> >
> > I have read your reply carefully. I think most of my questions are answered, and I am willing to increase my rating. However, before that I would suggest the author take a look at the following paper:
> > ======
> > Bao Wang, Binjie Yuan, Zuoqiang Shi, Stanley J. Osher. ResNets Ensemble via the Feynman-Kac Formalism to Improve Natural and Robust Accuracies, arXiv:1811.10745, NeurIPS, 2019
> > ======
> >
> > In the above paper, the authors model ResNet as a transport equation, which is different from the ODE models. Also, the authors interpret the adversarial vulnerability as the irregularity of the decision boundary. Finally, the authors introduce a diffusion term to enhance the robustness of the neural nets. The results show that the proposed method remarkably improves both robust and natural accuracies of the adversarially trained deep nets. I think it is a different way to improve the robustness of the ODE models, the authors should address this in your paper.

---

> > > ### Author Response · Authors · 2019-11-14
> > > **Thanks a lot for the reply (AnonReviewer1)**
> > >
> > > Thanks for providing this interesting work.  Interpreting the adversarial robustness from the view of the transport equation is really exciting. We have updated our revision and cited this work in the section of related works.
> > >
> > > Thanks.

---

> ### Author Response · Authors · 2019-11-13
> **Response (AnonReviewer1)  - Part 2**
>
> [### Second ###] Let us discuss the experiments on the comparison between CNN models and neural ODE-based models (Q.3/4/6).
>
> ==>> For Q.6, we conducted the experiments in which, both the number of parameters and the number of function evaluations are controlled to be the same for the neural ODE-based and CNN models. For the neural ODE-based models, the time range is set from 0 to 1. We use the Euler method, and the step size is set to be 0.05. Thus the number of evaluations is 1/0.05=20. For the CNN models (specifically ResNet), we repeatedly concatenate 20 residual blocks, and these 20 blocks share the same weights. We conduct experiments on the MNIST dataset, and all the models are trained only with original non-perturbed images. Our experiments show that, in this condition, the neural ODE-based models still outperform the CNN models (FGSM-0.15: 87.5% vs. 81.9%,   FGSM-0.3: 53.4% vs. 49.7%,   PGD-0.2: 11.8% vs 4.8%).
>
> 	Here, we also want to explain why we only control the number of parameters in our original submission. The reasons are as follows: [5] and [6] show that the residual block can be interpreted as the single-step discretization of a continuous ODE. Thus, from the view of an ODE, a residual block and a neural ODE both model the dynamics within the same time range 0 to 1, but with different step sizes. In this sense, if we compare two methods by matching the number of function evaluations of ODE with the depth of CNNs, the 20 weight-sharing residual blocks model the dynamics in the time range of 0-20, which is not consistent with the time range (0-1) of neural ODE method. Thus, it is reasonable to compare two kinds of models by controlling the same number of parameters and the same time range, instead of matching the number of function evaluations with the depth of CNN models.
>
> 	Besides, we elucidate another advantage of neural ODE here, namely, flexible computation. After the neural ODE-based models are well-trained, the number of evaluations (the steps of the Euler method) can be chosen to be smaller without too much loss of accuracy and robustness. During the test, we changed the step size of the aforementioned model from 0.05 to 0.1. Here are the performances to demonstrate robustness (FGSM-0.15: 86.1%, FGSM-0.3: 50.3%, PGD-0.2: 8.6%). These performance indices are still better than/comparable to the 20-step CNN models. From the view of engineering, one can also reduce the depth after training, but the modeling time range will also change. It is not reasonable and will lead to obvious performance degradation (FGSM-0.15: 76.3%, FGSM-0.3: 35.5%, PGD-0.2: 3.5%).
>
> ==>> For Q.4, we implemented the adversarial training for the comparison. We train models with FGSM examples (epsilon=0.3) on the MNIST dataset. The results are shown in the following table, and we can see that ODENets still perform better than CNN models, and the proposed TisODEs are more robust in comparison to vanilla neural ODEs.
> ——————————————————————————
> Acc(%)    | \sigma=100 | FGSM 0.3 | FGSM 0.5 | PGD 0.3
> ——————————————————————————
> CNN	|	58.0	          |	98.4	       |	21.1	    |	5.3
> ——————————————————————————
> ODENet |	84.2	          |	99.1	       |	36.0	    |	12.3
> ——————————————————————————
> TisODE  |	87.9	          |	99.1	       |	66.5	    |	78.9
> ——————————————————————————
>
> ==>> For Q.3. In our original submission, the three used datasets include images that contain digits(MNIST, SVHN) and natural images (ImgNet10, which is a subset of ImageNet). Thanks for the suggestion to conduct more experiments. During this discussion period, we did experiments on the CIFAR10. We trained all the models only with original non-perturbed images and evaluated the robustness of models to random Gaussian noise and FGSM adversarial attacks. The results are shown in the following table:
> —————————————————————————————-
> Acc(%) | \sigma=15 | \sigma=20 | FGSM 8/255 | FGSM 10/255
> —————————————————————————————-
> CNN       |	70.2	|	 57.6	|	 24.3	|	18.4
> —————————————————————————————-
> ODENet |	72.6	|	 60.6	|	 31.2	|	26.0
> —————————————————————————————
> TisODE  |	74.3	|	 62.0	|	 33.6	| 	26.8
> ————————————————————————————---
> From the table above, we can observe that the ODENet is more robust than the CNN model in terms of both the random noise and the FGSM attack. Besides, our proposal, TisODE, can further improve the robustness of the vanilla neural ODE.
> (Here, we control the number of parameters to be the same for all the models, which is the same setting used in our original submission. The network used is small, which consists of five convolutional layers and one linear layer. The architectures are as follows: convUnit = conv → GroupNorm → ReLU)
> CNN: convUnit → convUnit → convUnit → ResBlock( convUnit → convUnit) → adaptive average pooling → Linear
> ODE-based models: convUnit → convUnit → convUnit → ODEBlock( convUnit → convUnit) → adaptive average pooling → Linear

---

> ### Author Response · Authors · 2019-11-13
> **Response (AnonReviewer1) - Part 3**
>
> [### Third ###] On the understanding of the robustness of neural ODE-based models (Q1 and 5).
>
> ==>> For Q5, we mentioned the continuity conditions in the content of the paper (outside the theorem), but unfortunately, this is not stated within the statement of Theorem 1 itself. Thanks for pointing out this. We have made the theorem rigorous in the revised version by adding the conditions of global Lipschitz continuity in states and the continuity in time. Thanks again.
>
> ==> For Q1, the non-intersecting property is certainly not a sufficient condition, but we want to use to provide some insights to explain the robustness of neural ODEs. Given examples that are classified correctly, for a new example that is surrounded by these correctly classified ones, its integral curve could be regularized to follow the curves of given examples. We aim to provide a plausible argument to shed light on the robustness of neural ODEs, which however is still not so clear. This is also one of our future research topics.

---

> ### Author Response · Authors · 2019-11-13
> **Response (AnonReviewer1) - Part 4**
>
> [### Fourth ###]  At last, on the related works (Q7).
>
> Our work aims to study the robustness of the neural ODE. We empirically compare the neural ODE-based models to CNN models in terms of robustness to random noise and adversarial examples. Furthermore, we put effort into the improvement of the robustness of the vanilla neural ODE and propose the TisODE. Thus, in our experiments, we compared the TisODE with the vanilla version. Besides, we also showed that our proposed method works in conjunction with several popular defense methods [8, 9], instead of beating any other state-of-the-art methods.
>
> 	Thanks a lot for listing several papers on the robustness of neural networks. Three of them [2, 3, 4] improve the robustness of neural networks by replacing the output layers with novel interpolating functions. These works are interesting, and we will cite these papers as important references for robustness improvement. In the paper [1], the authors propose a novel two-step method to improve the natural and robust accuracies by injecting noise and averaging predictions. However, these works are not related to the study on neural ODEs, and we feel it is not necessary to compare our proposal with them, because the focus of the current work is on the study of the robustness of a specific architecture--neural ODEs--and possible enhancements to the vanilla architecture to further improve their robustness performance.
>
> Thanks.
>
> [1]. Bao Wang, Binjie Yuan, Zuoqiang Shi, Stanley J. Osher. ResNets Ensemble via the Feynman-Kac Formalism to Improve Natural and Robust Accuracies, arXiv:1811.10745, NeurIPS, 2019
>
> [2]. Bao Wang, Xiyang Luo, Zhen Li, Wei Zhu, Zuoqiang Shi, Stanley J. Osher. Deep Neural Nets with Interpolating Function as Output Activation, NeurIPS, 2018
>
> [3]. Bao Wang, Alex T. Lin, Zuoqiang Shi, Wei Zhu, Penghang Yin, Andrea L. Bertozzi, Stanley J. Osher. Adversarial Defense via Data Dependent Activation Function and Total Variation Minimization, arXiv:1809.08516, 2018
>
> [4]. B. Wang, S. Osher. Graph Interpolating Activation Improves Both Natural and Robust Accuracies in Data-Efficient Deep Learning, arXiv:1907.06800
>
> [5]. Tian Qi Chen, Yulia Rubanova, Jesse Bettencourt, and David K Duvenaud. Neural ordinary differential equations. In Advances in neural information processing systems, pp. 6571–6583, 2018.
>
> [6]. Yiping Lu, Aoxiao Zhong, Quanzheng Li, and Bin Dong. Beyond finite layer neural networks: Bridging deep architectures and numerical differential equations. arXiv preprint arXiv:1710.10121, 2017.
>
> [7]. Xu, Huan, and Shie Mannor. "Robustness and generalization." Machine learning 86.3 (2012): 391-423.
>
> [8]. Cihang Xie, Jianyu Wang, Zhishuai Zhang, Zhou Ren, and Alan Yuille. Mitigating adversarial effects through randomization. arXiv preprint arXiv:1711.01991, 2017.
>
> [9]. Cihang Xie, Yuxin Wu, Laurens van der Maaten, Alan L Yuille, and Kaiming He. Feature denoising for improving adversarial robustness. In Proceedings of the IEEE Conference on Computer Vision and Pattern Recognition, pp. 501–509, 2019.

---

### Official Review · AnonReviewer2 · 2019-10-23
**Official Blind Review #2**

**Rating:** 8

**Review:**

The paper is concerned with neural ODE-based networks, specifically their robustness. While ODEs are a classical subject in mathematics with many applications in the sciences and beyond, neural ODEs are a recently proposed family of models for nonlinear mappings in the context of machine learning systems. There they show promise and are an active field of research.

The paper makes two primary contributions. (1) It studies the robustness of neural ODE, and (2) proposes a more robust variant of neural ODEs. For (1), robustness to both perturbed and adversarial inputs is considered, and theoretical interpretations for the robustness of neural ODEs are given. These theoretical insights form the basis of the contribution in (2).

The paper is well and clearly written, supplies most of the necessary theoretical background and offers useful contributions. I recommend the paper for publication.

In terms of improving the paper further, I’d suggest a slightly less casual treatment of the conditions under which the mathematical statements quoted hold. E.g. Theorem 1 is part of the classical Picard-Lindelof theorem and requires similar conditions (or at least the conditions of the necessary and sufficient, but less well known, Okamura's theorem, see [1]). A differentiable counterexample if these conditions don’t hold can be found e.g. in Wikipedia [2]. I see that the authors have responded to that point on the openreview website. I’d suggest however that for a result going back to the early 19th century citing a paper from 2019 (which itself cites a textbook on computational anatomy) seems suboptimal from an educational point of view.

Another possible improvement of the paper could be to expand the adversarial attacks considered to the gradient-free optimization techniques employed in e.g. [3] which have sharply reduced other defenses against adversarial attacks.

[1] https://www.ams.org/journals/proc/1967-018-04/S0002-9939-1967-0212240-6/S0002-9939-1967-0212240-6.pdf
[2] https://en.wikipedia.org/wiki/Picard–Lindelöf_theorem#Example_of_non-uniqueness
[3] https://arxiv.org/abs/1802.05666

**Experience Assessment:**

I have read many papers in this area.

**Review Assessment: Checking Correctness Of Derivations And Theory:**

I carefully checked the derivations and theory.

**Review Assessment: Checking Correctness Of Experiments:**

I assessed the sensibility of the experiments.

**Review Assessment: Thoroughness In Paper Reading:**

I read the paper at least twice and used my best judgement in assessing the paper.

---

> ### Author Response · Authors · 2019-11-13
> **Response (AnonReviewer2)**
>
> Thanks a lot for the constructive review.
>
> 1. We agree that the mathematical statements  (Theorem 1) should be rigorous. We have modified the statement in the revised version. In particular, we add the conditions of global Lipschitz continuity in states and the continuity in time. In our original submission, we have cited a textbook [4] on ODEs to reference the theorem. In the revision, we also cite another well-known textbook [5]. Thanks.
>
> 2. Following the suggestion, we experimented with the gradient-free attack method (SPSA) in [3].  We evaluated the performance of models trained with Gaussian perturbations on the MNIST, by choosing n=50, T=10, and epsilon = 0.4 for the SPSA attack. The results show that the neural ODE-based models are more robust than  CNN models in front of such attack method. Besides, the proposed TisODE outperforms the vanilla neural ODE. (CNN: avg 33.4%; ODENe: avg 43.1%; TisODE: avg 45.3%)
> We also evaluated these models with a black-box attack method ( ZOO [6] with epsilon=0.4), which is also one type of gradient-free attacks. We still observed that neural ODE-based models are more robust than CNN models and the proposed TisODE enhances the robustness of the vanilla neural ODE. (CNN: avg 15.6%; ODENet: avg 51.0%; TisODE: avg 52.5%)
>
>
> [4] Laurent Younes. Shapes and diffeomorphisms, volume 171. Springer, 2010.
> [5] Coddington, Earl A., and Norman Levinson. Theory of ordinary differential equations. Tata McGraw-Hill Education, 1955.
> [6]Pin-Yu Chen, Huan Zhang, Yash Sharma, Jinfeng Yi, and Cho-Jui Hsieh. Zoo: Zeroth order optimization based black-box attacks to deep neural networks without training substitute models. In Proceedings of the 10th ACM Workshop on Artificial Intelligence and Security, pp. 15–26. ACM,2017.

---

### Official Review · AnonReviewer3 · 2019-10-25
**Official Blind Review #3**

**Rating:** 6

**Review:**

This paper investigates the robustness of Neural Ordinary differential equations (ODEs) against corrupted and adversarial examples. The crux of the analysis is based on the separation property of ODE integral curves. The insights from empirical robustness evaluation show that controlling the difference between neighboring integral curves is able to improve neural ODE's robustness. In general, neural ODE is a hot research topic in recent years, and a paper advancing knowledge in this area about understanding its various characteristics is certainly welcome. The paper is well motivated and clearly written. One aspect that confuses me a little originally is the different effects of getting ridding of the dependency on the time t and adding the steady state regularization. It would be nice to elucidate which part makes more contributions? Furthermore, to compare the robustness of the new approach with CNN, the input data consists of original images and their Gaussian-noise based perturbed samples. Since the paper already involves the evaluation using adversarial examples, it will make the paper much more stronger to show that when training both the new approach and the CNN with adversarial training, the proposed regularization can still lead to better robustness.

**Experience Assessment:**

I have read many papers in this area.

**Review Assessment: Checking Correctness Of Derivations And Theory:**

I assessed the sensibility of the derivations and theory.

**Review Assessment: Checking Correctness Of Experiments:**

I assessed the sensibility of the experiments.

**Review Assessment: Thoroughness In Paper Reading:**

I read the paper at least twice and used my best judgement in assessing the paper.

---

> ### Author Response · Authors · 2019-11-13
> **Response (AnonReviewer3)**
>
> Thanks a lot for the review and the suggestions.
>
> 1. With regard to the time-invariant property and the steady-state constraint, the steady-state constraint enhances the robustness of vanilla neural ODEs based on the time-invariant property. To wit, consider a certain integral curve z1 and its neighboring curve \tilde_z1, and assume \tilde_z1(0)=z1(T’). Without the time-invariant property,  Eqn. (3) does not hold. Consequently, the steady-state constraint cannot control the difference between the two states at time t1 only with the information of curve z1. Thus,  these two parts cannot be separated. As such, we feel that it is not necessary to perform an ablation study on these two aspects.
>
> 2. Thanks for the advice of evaluating models with adversarial training. We implemented the adversarial training of the models on the MNIST dataset, and the adversarial examples for training are generated in real-time via the FGSM method  (epsilon=0.3) during each epoch [1]. The results of the adversarially trained models are shown in the following table:
> —————————————————————————————
> Acc(%)      | \sigma=100 | FGSM 0.3 | FGSM 0.5  | PGD 0.3
> —————————————————————————————
> CNN	  |	58.0	           |	98.4	       |	21.1	      |	5.3
> —————————————————————————————
> ODENet    |	84.2	           |	99.1	       |	36.0	      |	12.3
> —————————————————————————————
> TisODE     |	87.9	           |	99.1	       |	66.5	      |	78.9
> ————————————————————————————---
> We can see that the neural ODE-based models are consistently more robust than CNN models. Besides, the proposed TisODE also outperforms the vanilla neural ODE. We have added these experiments to the revision (Appendix setion 7.4).
>
> [1] Aleksander Madry, Aleksandar Makelov, Ludwig Schmidt, Dimitris Tsipras, and Adrian Vladu. Towards deep learning models resistant to adversarial attacks. arXiv preprint arXiv:1706.06083, 2017.

---

### Public Comment · ~Hongyang_Zhang1 · 2019-10-15
**How many iterations in the PGD attack?**

Thanks for the interesting work.

I am wondering how many iterations of the PGD attacks do you use in your experiments? According to my own experiments, neural ODE which is trained on unperturbed images might not be very robust to PGD attack with many iterations, like PGD with1,000 iterations. Have you observed a similar phenomenon? I think it is critical to evaluate the method by the "worst-case" attacks.

---

> ### Author Response · Authors · 2019-10-16
> **Response**
>
> Thanks for the comment. Here, we use 10 iterations of the PGD attack and apply the same adversarial setting among all models to show the difference in robustness.
>
> PGD is a strong attacker for evaluating the robustness of a model. In [1], the authors evaluate adversarially trained models by using 7, 20, 40 iterations PGD on the MNIST and CIFAR10 datasets. However, in our paper, we do not train any models with adversarial examples, but all models are trained only on original images together with their Gaussian perturbed versions. So, we think it is reasonable to use 10 iterations here.
>
> Besides, we also tried applying the PGD attack with a larger epsilon or more iterations. In this case, the PGD attacker is very easy to totally mislead all models except the proposed TisODE, because these models are not adversarially trained. So, the 1000-iterations PGD maybe not a good choice to compare the robustness of different models in this paper.
>
> In terms of your concern, we just ran 100-iteration PGD attack on the MNIST dataset. The neural ODE-based models are consistently more robust in comparison to CNN models (57.0% -vs- 18.3%).
>
> [1] Aleksander Madry, Aleksandar Makelov, Ludwig Schmidt, Dimitris Tsipras, and Adrian Vladu. Towards deep learning models resistant to adversarial attacks. arXiv preprint arXiv:1706.06083, 2017.

---

> > ### Public Comment · ~Anthony_Wittmer1 · 2019-10-16
> > **On evaluating robustness**
> >
> >  Sorry, I do not agree that it's a little odd to use strong attack in this setting.
> >
> > When evaluating the robustness of a new model, the goal is to show whether the model is robust against _all_ possible attacks within a threat model. For instance, the robustness to FGSM attack does not constitute progress if the model is vulnerable to PGD attack. This is why the robust accuracy of a model is defined as the *minimum* accuracy achieved against the worst-case attack within the threat model.
> >
> > In addition, besides the adversarial robustness, the authors could consider another evaluation of robustness, the robustness on the common corruption, such as the dataset CIFAR-C and ImageNet-C.

---

> > > ### Author Response · Authors · 2019-10-21
> > > **Response**
> > >
> > > Thanks for the comments. One of the main objectives of this work is to investigate the robustness of neural ODE-based models in comparison to CNN models. To achieve this goal, we use the same kind of perturbation to evaluate the robustness of neural ODE-based models and CNN models. The perturbations include random Gaussian noise and harmful adversarial examples,  such as FGSM and PGD attack.
> > >
> > > For the first concern, in work [1], the authors use PGD to evaluate the adversarially trained models (40-iterations on MNIST and 7/20 on CIFAR). Thus, in our work, for the models without adversarial training, it is reasonable to use a 10-iteration PGD attack for comparing the robustness of neural ODE-based models with their CNN counterparts. Besides, we tried the 100-iteration PGD attack. As mentioned in the answer above,  the robustness of neural ODE-based models consistently outperforms CNN models.
> > >
> > > For the other concern on the common corruption, in fact, we evaluated the robustness of all the models in terms of random Gaussian noise, which is a ubiquitous form of perturbation.

---

### Public Comment · ~Chen_Liu1 · 2019-10-16
**An interesting work & some confusions**

Thanks for the very interesting work combining adversarial robustness with Neural ODE. I enjoy reading it : )

I have some confusions:

1. Since neural ODE can be considered as a generalization of ResNet to continuous layer depth, isn't it a bit "unfair" to compare neural ODE with CNN instead of ResNet?
2. Did you miss some assumptions of Theorem 1, like the smoothness of function f? Because for example if we set f(z(t), t) to be 1 if z(t) = t and 0 otherwise, it seems that Theorem 1 is not true.
3.  The robustness against an adversarial attack means the lack of adversary in the entire adversarial budget, but the analysis in Section 4.1 cannot rule out the perturbation to the points which are not in z1 trajectory but in the adversarial budget.

---

> ### Author Response · Authors · 2019-10-17
> **Response**
>
> Thanks a lot for the comments.
>
> Question 1: In this work, the CNN models are constructed in ResNet architecture.  The part of representation mapping (RM) in each CNN model consists of a residual block(s).  It could be better to state this explicitly in the main article. Thanks.
>
> Question 2:  In the neural ODE, the nonlinear mapping from input to output is modeled by a continuous-time ODE.  It is right the function f should be continuous in time t and globally Lipschitz continuous in state z.  The proof of Theorem 1 can be found in [2] (Appendix A.1).
>
> Question 3: We appreciate this comment. In section 4.1, we consider the perturbations that are also on the trajectory of a certain point.  A robust model should accurately handle these neighboring points.  Thus, the steady-state constraint on these points is a necessary condition for the robustness. Although this constraint does not include all the neighboring points,  it still can contribute to the improvement of robustness.
>
>
> [2] Emilien Dupont, Arnaud Doucet, and Yee Whye Teh. Augmented neural odes. arXiv preprint arXiv:1904.01681, 2019.

---

### Public Comment · ~Yiping_Lu1 · 2019-12-23
**Some related works**

Congrats to the very interesting work on the robustness of the ODE-Nets. This work inspires me a lot.

But I found out that it seems that your manuscript  is missing some important references
Zhang J, Han B, Wynter L, et al. Towards robust resnet: A small step but a giant leap[J]. arXiv preprint arXiv:1902.10887, 2019.

In my understanding, Euler scheme is x = x+ stepsize*rhs function
There approach is letting stepsize to be samller and your approach  is to let the rhs function smaller, what's the connection and difference. Which one is better?

Our work
Dinghuai Zhang*, Tianyuan Zhang*,Yiping Lu*, Zhanxing Zhu, Bin Dong. "You Only Propagate Once: Painless Adversarial Training Using Maximal Principle." (*equal contribution) 33rd Annual Conference on Neural Information Processing Systems 2019(NeurIPS2019).
is also a method to train a robust ODE
The reason why we formulate the problem as the setting in our paper is that we want our method can applied to any neural architecture. My question is that
How your analysis can inspire algorithms to train a robust model of any architecture ?

---

> ### Author Response · Authors · 2019-12-26
> **Thanks for the interest and for providing two amazing works.**
>
> For the work by J Zhang et al., it claims a small step size 'h' can benefit the robustness of a deep network. This conclusion is drawn from Proposition 1 and 2. Usually, the number of layers 'D' is fixed, then changing the step size 'h' also leads to the change of the time range modeled (T=h*D).  Thus, this work may be regarded as how to choose a proper time range 'T' given a network with a fixed number of layers. In the last two paragraphs of Section 3, the authors also mention that a deeper ResNet requires a smaller 'h,' but a shallower ResNet allows for a larger 'h.'  I just guess that, even for a shallow network, the step size should not be too small, it may influence the approximation capability. In Figure 9 of this work, the performance of the green line (depth=20) drops obviously when 'h' is very small.
>
> In our work, we fix the time range modeled (0 -to- T). The improvement includes introducing the time-invariant property and adding a steady constraint outside the modeling time range (T -to- 2T).  I think It is hard to make a comparison between these two works.
>
> Also thanks for providing the YOPO work, and I am very interested in it. This method can be applied to train any neural network; it is really cool. For neural ODEs, engineers can simply replace a res-block with a neural ODE block, which is also mentioned by David Duvenaud in his NeurIPS talk.  Then, training the neural ODE-based networks with our proposed method can yield a robust model. BTW, I really appreciate your works that connect dynamical systems with deep learning. Looking forward to more discussion.   --- HS

---

> > ### Public Comment · ~Yiping_Lu1 · 2020-02-04
> > **Changing the scaling**
> >
> > Ok I can reformulate my question as
> > For an ODE model \dot X = f(X,t)
> > - using small step size [this is zhang et al's approach
> > - let f(X,t) smaller [this is your approach
> > What if I reformulate the problem as following:
> > - first t' = t/a  here a is a scaling factor
> > - \dot X = af(X,t')
> > then using the same step size \Delta t to discrete
> > is equivalent to using small step size to  discrete the original ode
> >
> > Yes, the number of layers 'D' is fixed, the tricky thing is that you can let the time of the ode model to be [0,1] also can be [0,2]. The scaling don't affect.
> >
> > Thus I prefer to directly analysis the discrete neural network, the lip constant is just \pi_i (1+stepsize*||f_i||)...

---

### Decision · Program_Chairs · 2019-12-19

**Decision:**

Accept (Spotlight)

**Comment:**

This paper studies the robustness of NeuralODE, as well as propose a new variant. The results suggest that the neuralODE can be used as a building block to build robust deep networks. The reviewers agree that this is a good paper for ICLR, and based on their recommendation I suggest to accept this paper.